∂ | **Open Peer Review** | Host-Microbial Interactions | Research Article

# SARS-CoV-2 infection severity and mortality is modulated by repeat-mediated regulation of alternative splicing

Priyanka Mehta,[1,2] Partha Chattopadhyay,[1,2] Varsha Ravi,[1] Bansidhar Tarai,[3] Sandeep Budhiraja,[3] Rajesh Pandey[1,2]

**ABSTRACT**  Like single-stranded RNA viruses, SARS-CoV-2 hijacks the host transcriptional machinery for its own replication. Numerous traditional differential gene expression-based investigations have examined the diverse clinical symptoms caused by SARS-CoV-2 infection. The virus, on the other hand, also affects the host splicing machinery, causing host transcriptional dysregulation, which can lead to diverse clinical outcomes. Hence, in this study, we performed host transcriptome sequencing of 125 hospital-admitted COVID-19 patients to understand the transcriptomic differences between the severity sub-phenotypes of mild, moderate, severe, and mortality. We performed transcript-level differential expression analysis, investigated differential isoform usage, looked at the splicing patterns within the differentially expressed transcripts (DET), and elucidated the possible genome regulatory features. Our DTE analysis showed evidence of diminished transcript length and diversity as well as altered promoter site usage in the differentially expressed protein-coding transcripts in the COVID-19 mortality patients. We also investigated the potential mechanisms driving the alternate splicing and discovered a compelling differential enrichment of repeats in the promoter region and a specific enrichment of SINE (Alu) near the splicing sites of differentially expressed transcripts. These findings suggested a repeat-mediated plausible regulation of alternative splicing as a potential modulator of COVID-19 disease severity. In this work, we emphasize the role of scarcely elucidated functional role of alternative splicing in influencing COVID-19 disease severity sub-phenotypes, clinical outcomes, and its putative mechanism.

**IMPORTANCE**  The wide range of clinical symptoms reported during the COVID-19 pandemic inherently highlights the numerous factors that influence the progression and prognosis of SARS-CoV-2 infection. While several studies have investigated the host response and discovered immunological dysregulation during severe infection, most of them have the common theme of focusing only up to the gene level. Viruses, especially RNA viruses, are renowned for hijacking the host splicing machinery for their own proliferation, which inadvertently puts pressure on the host transcriptome, exposing another side of the host response to the pathogen challenge. Therefore, in this study, we examine host response at the transcript-level to discover a transcriptional difference that culminates in differential gene-level expression. Importantly, this study highlights diminished transcript diversity and possible regulation of transcription by differentially abundant repeat elements near the promoter region and splicing sites in COVID-19 mortality patients, which together with differentially expressed isoforms hold the potential to elaborate disease severity and outcome.

**KEYWORDS**  COVID-19, differential transcript expression, alternative splicing, alternate promoter usage, repeat elements

Address correspondence to Rajesh Pandey, rajeshp@igib.in.

Priyanka Mehta and Partha Chattopadhyay contributed equally to this work. Author order was determined based on scientific contribution to the study and in concurrence with the corresponding author.

The authors declare no conflict of interest.

See the funding table on p. 21.

The COVID-19 pandemic disrupted not only the daily human life, but also the cellular machinery within the infected cells. SARS-CoV-2 is a single-stranded RNA virus with extremely diverse clinical manifestations ranging from asymptomatic infections to acute respiratory distress necessitating the use of supplemental oxygen (1). A slew of recent studies, using next-generation sequencing, have shed light on how the host responds to this invading pathogen challenge (2–5). However, a common theme of these studies has been to explore the differential gene expression patterns to elucidate the host response. Although changes at gene level are important for understanding the biological processes disrupted within the cell, the cell produces multiple different isoforms of the same gene by a process known as alternative splicing (AS). It is a key phenomenon which increases the mature transcript diversity as well as generates protein complexity/specificity within a cell. Alternative splicing has significant functional implications as it changes the proteins encoded by the mature transcripts (6). Consequently, this process is governed by a complex yet dynamic mechanism involving numerous *cis*-acting elements and *trans*-acting factors that are guided by the functional coupling of transcription and splicing.

Infectious agents such as DNA and RNA viruses (Herpes Simplex Virus-1, Zika, Dengue, Influenza A, and SARS-CoV-2) as well as bacteria (*Mycobacterium tuberculosis*) are known to hijack the host transcriptional machinery post infection for their own replication, leading to changes in the host splicing machinery (7–12). These infectious agents can directly regulate host gene expressions, e.g., dengue viral protein NS5 interacts with the snRNP component of the spliceosome to alter the host splicing machinery (9). Alternatively, viruses can cause accumulation of splicing factors within the nucleus resulting in changes in the concentration of proteins, thus contributing to the alternative splicing of host genes (13). Another well studied infectious agent *Mtb* is known to reprogram the transcriptional profile of macrophages upon infection (12). Similarly, the SARS-CoV-2 Nsp16 is also proposed to bind to snRNAs and cause host splicing changes (11). While these studies emphasize the significance of alternative splicing in determining host cellular responses during infection, the clinical implications of such changes are yet to be thoroughly investigated.

The primary aim of this study is to understand modulators of disease severity sub-phenotypes within patients albeit infected by the similar/same pathogen. Therefore, in this study, we compared the host transcriptomic signatures of 125 RT-PCR positive and sequencing-confirmed SARS-CoV-2 hospital admitted patients with varying disease outcomes, ranging from mild, moderate, and severe to fatality. Toward that, we explored and elucidated the potential role of differential transcript isoform expressions between mild, moderate, severe, and mortality patients in modulating the COVID-19 disease severity. We further aimed to understand the possible mechanisms of transcription regulation via transcription factors as well as differential distribution of repeat elements such as SINEs (Alu and MIR), LINEs (L1 and L2), and LTRs underpinning the differentially expressed transcripts.

## RESULTS

### Patient cohort characterization, classification, and clinical evaluation

We recruited 125 hospitalized COVID-19 patients to understand the dynamics of alternative transcription and splicing and how/whether they can modulate the disease severity. We stratified the patients into four groups based on the disease severity and clinical outcome: mild, moderate, severe, and mortality, as per the COVID-19 disease severity indices by the Indian Council of Medical Research (ICMR). We performed transcriptome sequencing of the 125 patients from the nasopharyngeal swabs collected at the time of hospitalization, followed by transcript-level RNA-seq data analysis to identify differential transcript expression, differential transcript isoform usage, and splicing pattern across the COVID-19 severity sub-phenotypes (Fig. 1A).

The demographic and clinical data of the individuals are presented in the Table S1. The median age of the mild patients was significantly lower compared to the moderate

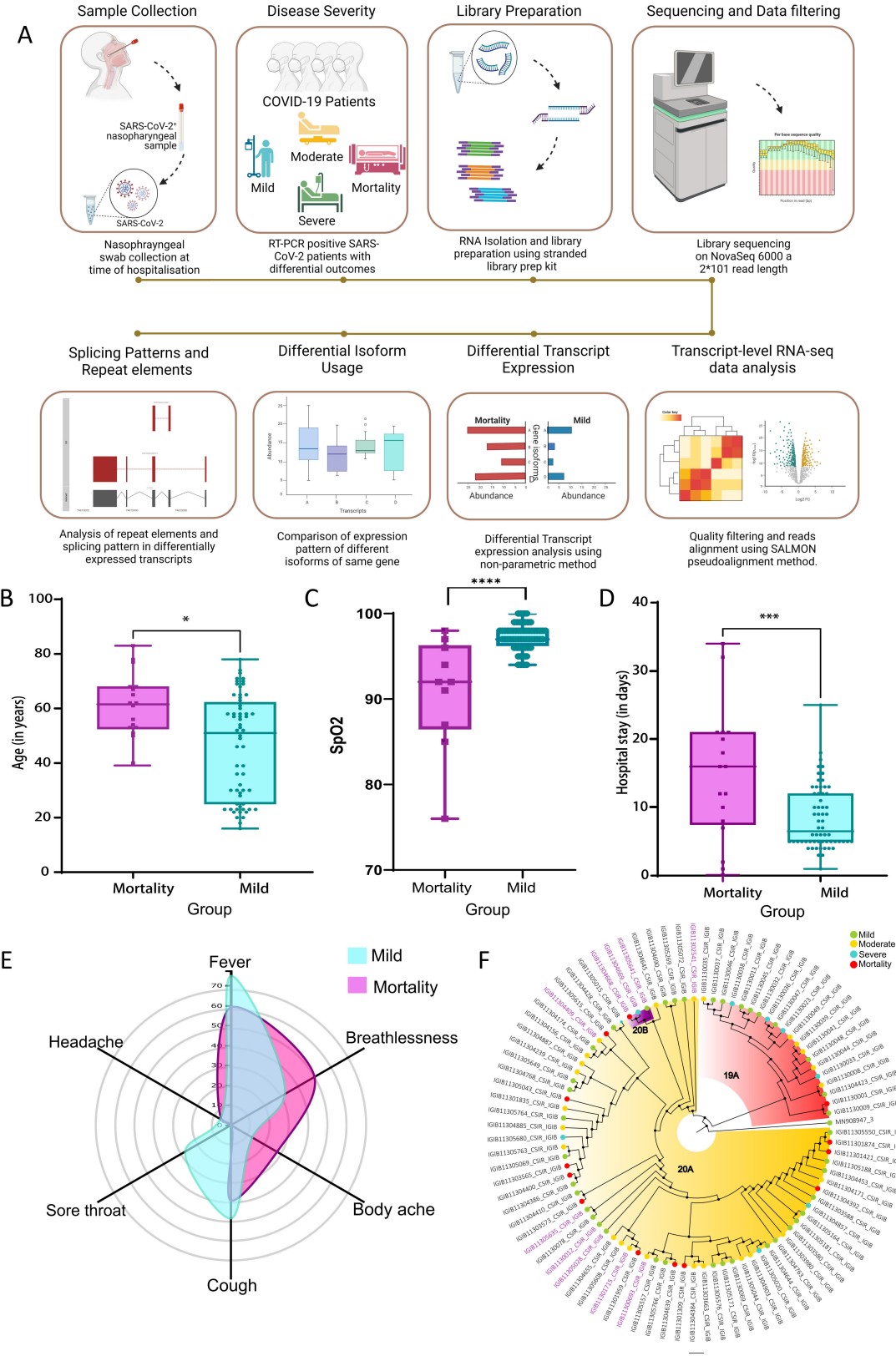

**FIG 1** Overview of study design, patient segregation with clinical characterization. (A) Sample distribution and schematic workflow for transcriptome sequencing, transcript-level RNA-Seq data analysis, downstream functional analysis and interpretation. (B to D) Sample-wise distribution of the clinical parameters across the mild and the mortality patients for (B) Age of the patients, (C) their SpO$_2$ (in %) levels, and (D) duration of their hospital stay. Upper

**FIG 1** (Continued)

bar is showing the statistical significance. (E) Diversity of symptoms presented by the mild and the mortality patients. Data are represented as the number of reported events, normalized to the number of patients in that group. *$P$-value < 0.05, **$P$-value < 0.01, ***$P$-value < 0.001, ****$P$-value < 0.0001. (F) Phylogenetic tree of the SARS-CoV-2 clades from the positive patients. Sample labeled with pink color belong to 20B. Disease severity types and SARS-CoV-2 lineages are distributed across the phylogeny as represented by the color of nodes (green for mild, yellow for moderate, blue for severe, and red for mortality).

($P$ value 0.006), severe ($P$ value 0.015), and mortality patients ($P$ value 0.0379) (Fig. S1A; Fig. 1B). We observed a significantly higher $SpO_2$ ($P$ value < 0.001) and shorter duration of hospital stay in the mild group compared to the mortality ($P$ value 0.008) (Fig. 1C and D). The $SpO_2$ was significantly higher in the moderate group as compared to the severe ($P$ value 0.0005) and mortality ($P$ value 0.017) groups (Fig. S1B). More mortality patients experienced breathlessness than the mild patients (Fig. 1E). Fever was a common symptom among all the patient sub-groups. Clinical data highlighted that both the moderate and the severe group patients frequently suffered from breathlessness when compared to the mild patients (Table S1).

The Ct value of the SARS-CoV-2 *E* and *RdRp* gene was also significantly lower in the mortality compared to the moderate patients ($P$ value 0.009) (Fig. S1C and D). A higher SARS-CoV-2 viral gene reads (M, N, S, and ORF1a gene) were detected in the moderate and severe patients compared to the mild (Fig. S1E; Table S2). We also sequenced the whole genome of the SARS-CoV-2 virus isolated from nasopharyngeal swabs of the patients to determine whether patients with varying severity levels are infected with different strains of the virus. Despite the differences in clinical severity and outcome, we discovered that the virus strain (19A, 20A, and 20B) was similarly distributed between the mild, moderate, severe, and mortality patients (Fig. 1F). Overall, these clinical, sequencing, and demographic data represent the diversity of symptoms within the COVID-19 sub-phenotypes despite similarity in the underlying viral infection and emphasize the need of understanding the transcriptional dynamics within the COVID-19 severity sub-phenotypes.

## Decreased transcript diversity associated with COVID-19 mortality

To understand the host transcriptional response within the COVID-19 sub-phenotypes, we first performed differential gene expression analysis between the mild/moderate/severe and the mortality patients. We identified 43 genes to be differentially expressed (DE) between the mortality and mild patients out of which 30 genes were downregulated in mortality (Fig. 2A; Table S3). In the mortality vs moderate, 3/7 differentially expressed genes (DEGs) were downregulated in mortality group, while in the mortality vs severe comparison, 22/24 genes were downregulated in the mortality patients (Fig. S1F and G; Table S3). Around 85% of the human genes is reported to have more than one transcript, and the dynamic expression of specific transcripts in the health and disease spectrum is reported to have a role in modulating disease trajectory (14). Therefore, we looked at the total transcripts available for the DE genes and the number of isoforms expressed (total expressed transcripts) in each COVID-19 disease sub-phenotype. Interestingly, when comparing mortality to the mild patients, we found a decrease in the number of expressed transcripts relative to the available transcripts (Fig. 2A). Notably, we observed a similar trend in the transcript expression in the mortality patients when compared to the moderate and the severe patients (Fig. S1H and I). To assess whether this is due to the overall suppression of the transcription process, we checked for expression of three human housekeeping genes, *ACTB* ($P$ value 0.59), *GAPDH* ($P$ value 0.33), and *TUBB* ($P$ value 0.71) between the mild and mortality groups. We observed no significant change in the expression of these three housekeeping genes between the mild and mortality, suggesting absence of global transcription suppression in the mortality patients (Fig. 2B to D). We further checked the cell type abundance in the two groups, which might have contributed to the different transcript diversity. Apart from the epithelial cells, we have identified several immune cells and few non-immune, non-epithelial cells, and none of them were significantly different between the mild and

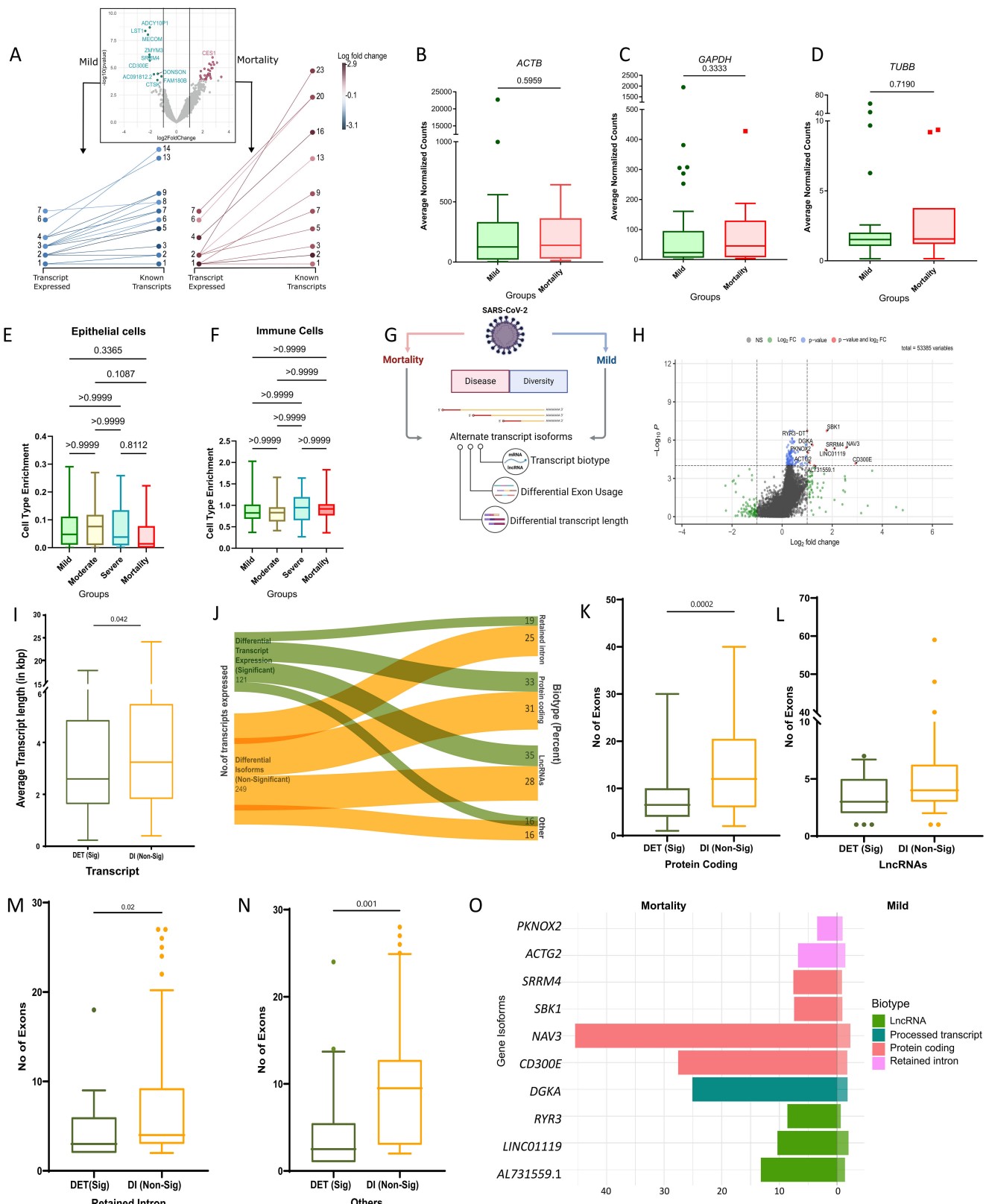

**FIG 2** Characterization of the transcript diversity and differentially expressed transcripts across mild and mortality patients. (A) Transcript diversity between the mild and mortality patients, represented as the number of transcripts expressed vs number of total transcripts of the DE genes. Multiple genes having the same number of transcripts/expressing the same number of transcripts are clubbed together. (B) Distribution of housekeeping genes *ACTB*, (C) *GAPDH*, and (Continued on next page)

**FIG 2** (Continued)

(D) *TUBB* between the mild and mortality patients. (E) Epithelial and (F) Immune cells distribution across the clinical sub-groups. (G) Graphical representation of the characterization of the transcript level diversity between the mild and the mortality. (H) Transcript level differential expression between the mild and the mortality. (I) Average transcript length between the differentially expressed transcripts (DTEs) and the transcript isoforms (DIs). (J) Biotype classification of the DTEs and DIs as per Ensembl. Number shows percentage of transcripts in each biotype. (K to N) Differential number of exons between the DTEs and DIs for (K) Protein coding transcripts, (L) LncRNA transcripts, (M) Retained introns, and (N) all other transcript biotypes. (O) Average abundance of 10 significantly differentially expressed transcripts between the mild, mortality, and their biotypes.

mortality patients (Fig. 2E and F; Fig. S1J and R). Thus, we observe that in the mortality patients, transcript diversity is decreased during an early host response to the COVID-19.

We then performed the transcript-level DE analysis to capture the transcript-level diversity with respect to the transcript biotype, differential exon usage, and differential transcript length across the sub-phenotypes (Fig. 2G). We identified 121 differentially expressed transcripts (DETs) in the mortality, compared to the mild, out of which 10 were significantly differentially expressed ($P$-adj ≤ 0.05, log2FC ≥ ± 1.5), while 111 were significantly different in the mortality patients based on only $P$-adjusted values ($P$-adj ≤ 0.05) (Fig. 2H; Table S4). We also identified a total of 249 isoforms of the 121 DTEs to be expressed but their expression was not significantly different (hereafter referred to as differential isoforms: DI) between the mild and mortality. Importantly, we found the average transcript length of the DTE to be significantly shorter than their non-significant counterparts, i.e., DI ($P$ value 0.042) (Fig. 2I). Therefore, we segregated both the DTEs, and the DIs based on the transcript biotype, as per Ensembl.

Majority of the transcripts were either protein-coding or non-protein coding transcripts, and a fraction of transcripts had retained their specific intron (Fig. 2J). Biotype groups apart from these three categories (such as processed transcripts, non-sense mediated decay) were clubbed together as "Others." Interestingly, we observed a significantly lesser number of exons within the DTE compared to the DI for protein coding transcripts ($P$ value 0.0004), however a similar trend was observed across other transcript biotypes (Fig. 2K to N). Taken together, we observed a suppression of alternative transcription processes evidenced by the lesser transcript diversity and shorter length transcripts with a comparatively smaller number of exons in the mortality patients compared to the mild.

We also performed transcript-level DE analysis between the moderate/severe and the mortality (Fig. S1S and T). Although the transcript expression was not significantly different between the moderate/severe and mortality, interestingly all the 10 transcripts significantly upregulated in the mortality (vs mild) were also upregulated in the mortality compared to the moderate and severe (Fig. 2O; Fig. S1U). Within the mortality patients, we compared the expression of these 10 transcripts to check if there is any association between the age of the patients and the outcome. While the average expression of these transcripts (excluding pseudogene AL731559.1) were more in the mortality patients above the median age of 61 compared to below (Fig. S2A), there was no significant association between the age and expression of these transcripts. This indicates that these transcripts are possibly associated with COVID-19 mortality cases; therefore, understanding the biological relevance of the differentially expressed transcripts is important to understand their role in modulating the COVID-19 disease severity.

## Alternative transcripts of protein coding genes in COVID-19 severity

To derive mechanistic insights into the differential transcripts, we performed pathway enrichment as well as gene set enrichment analysis for the 121 differentially expressed transcripts (Fig. 3A). We observed activation of metabolism and cellular organelle organization associated pathways in the mortality patients (Fig. 3B). Activation of organic and macromolecule metabolism is a common consequence of many virus infections, where the virus tends to induce high glucose metabolism and alter lipid metabolism for its replication (15, 16). Viruses also use actin cytoskeleton and microtubules for entry to the cell, as well as short- and long-range transport (17). At the same time, we

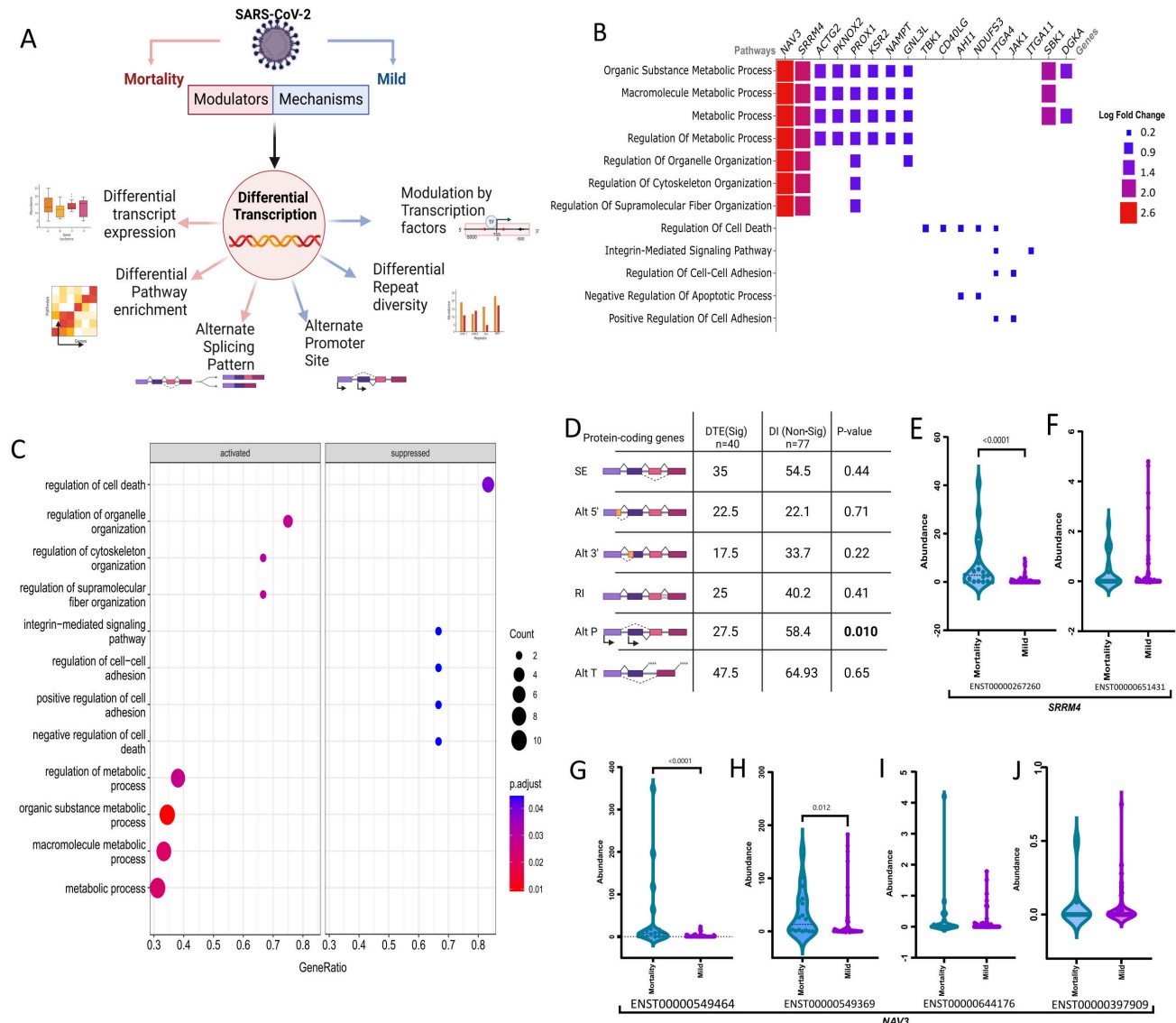

**FIG 3** Pathway enrichment analysis and types of alternative splicing mechanism observed. (A) Graphical representation highlighting the differential outcomes of SARS-CoV-2 infection, how alternative transcription modulates the outcomes, and possible mechanism of the modulation. (B) Pathway enrichment analysis of the 121 differentially expressed transcripts (DTEs). (C) Enrichment of genes within the pathways and expression of those genes. (D) Types of alternative splicing mechanisms observed in protein-coding transcripts: SE, Skipped Exon; Alt 5′, alternate 5′; Alt 3′, Alternate 3′; RI, retained intron; Alt P, Alternate promoter; Alt T, Alternate terminator. (E and F) Transcript specific expression of *SRRM4* and (G to J) *NAV3* between the mild and the mortality. DI, differential isoforms.

observed suppression of cell adhesion associated pathways in the mortality patients (Fig. 3B). While the gene set enrichment analysis reflected similar suppression of integrin cell surface interactions and extracellular matrix organization in the mortality patients, we also observe a positive enrichment of innate immune-related pathways (Fig. S2B). Multiple studies, including our previous study, have highlighted the role of cellular integrity in infection, and that dysregulation of cellular integrity aggravates the infection (18, 19). The apoptotic pathways were suppressed in the mortality patients, suggesting possible subversion of apoptosis by the virus in order to evade the host immune system (Fig. 3B) (20). Notably, we observed an overall higher association of protein coding transcripts, especially *NAV3* and *SRRM4*, with the metabolic process and cytoskeleton organization associated pathways, suggesting a significant role of protein coding transcript isoforms in the COVID-19 pathogenesis (Fig. 3C). Thus, coordinated

expression of these pathways might contribute to the disease severity in the COVID-19 mortality patients.

## Alternate promoter and termination site usage in the alternatively spliced transcripts

As the protein coding transcripts can modulate infection, we wanted to understand the splicing pattern and possible factors associated with the protein-coding transcripts (Fig. 3A). We used the SpliceDetector tool to predict the types of alternative splicing mechanisms observed in the DTEs and the DIs (21). Principally, exon skipping (SE) is the most prevalent type of alternative splicing event in the human genome (22). However, we observed alternate termination site usage as the major splicing pattern in the protein coding DTEs (Fig. 3D). The alternate promoter usage event was high in the DTE, albeit significantly less compared to the DIs. Interestingly, the retained intron and processed transcript biotypes also showed a high number of alternate promoter and termination splicing events in the DTE compared to the DI (Fig. S2C and D). In a recent study, Huin et al. linked alternate promoter usage with generation of shorter transcripts during the disease condition (23). Therefore, higher usage of alternate termination site and alternate promoter site in the DTEs could possibly explain the shorter transcript length and a smaller number of exons in the DTE. Since *NAV3* and *SRRM4* were significantly associated with infection, we looked at the transcript diversity of these two genes and found one out of the two expressed *SRRM4* transcripts and two out of the four expressed *NAV3* transcripts to be significantly differentially expressed in the mortality patients compared to the mild (Fig. 3E to J). *SRRM4* is a known regulator of alternative splicing, while *NAV3* is reported to be involved in multiple immune and infection-associated pathways such as FoxO signaling, T cell activation, and Human papillomavirus infection pathways (24, 25). Therefore, we propose that the promoter region may play an important role in shaping the alternative transcription *vis-à-vis* disease severity in COVID-19 disease.

## Transcription factors within repeats at the promoter region: possible modulator of Alternative Splicing

Two major mechanisms of promoter-mediated transcription regulation are through binding of transcription factors (TFs) and differential abundance of repeat elements within the promoter region of the gene (Fig. 4A). Multiple studies have reported enrichment dynamics of repeat elements in the upstream promoter region and its role in transcriptional regulation (26, 27). Notably, we also observed higher abundance of repeat elements within the promoter region of the significant *NAV3* and *SRRM4* transcripts expressed in the mortality patients when compared to their non-significant isoforms (Fig. S2E and F). Therefore, to understand the possible mechanisms of promoter site-mediated alternative splicing, we looked at the TF modulation and differential repeat element abundance between the DTEs and the DIs (Fig. 4A). We selected the 10 significantly differentially expressed transcripts to find the associated TFs using experimental database ChEA3 (28). We excluded one novel transcript AL731559.1, for which no data were available. We selected the top TFs based on their ranking from the ChEA3 database to have binding sites within the promoter region of these transcripts, some of which (CTCF, ZNF814, and ZFHX4) are predicted to regulate transcription by RNA polymerase II (Fig. 4B).

We selected a range of 5,000 bps upstream and 500 bps downstream of the transcription start site (TSS) as promoter region for repeat element enrichment dynamics analysis using RepeatMasker (Fig. 4C; Table S5). We calculated genome-wide repeat elements distribution in the selected promoter region for all known protein-coding and non-protein coding transcripts in the human genome to determine null distribution of repeats in the promoter region ($N$ = 131,038). Interestingly, abundance of short interspersed nuclear elements, SINEs (both Alu and mammalian-wide interspersed repeats, MIRs), long interspersed nuclear elements, LINE (L2), and long interspersed repeats, LTR elements within the promoter region of significantly differential transcripts

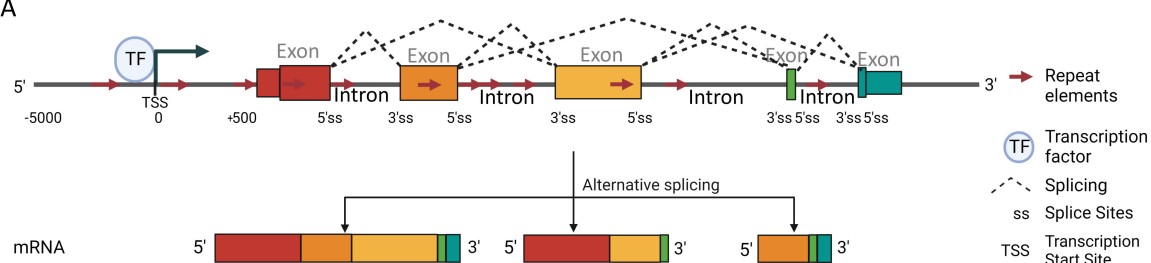

FIG 4 Binding of transcription factors and abundance dynamics of repeat elements with the promoter region of differentially expressed transcripts. (A) Schematic representation of possible modulators of alternate splicing. The transcription factor binding and repeat element abundance at the promoter region as well as within the transcript body as possible mechanisms. (B) Enrichment of TFs within the promoter region of the differentially expressed transcripts (Continued on next page)

**FIG 4** (Continued)

(DTEs) from the experimental ChEA3 database. (C) Graphical representation of the promoter region selected for repeat element abundance. (D) Differential Repeat element abundance within the promoter region of the DTEs and the differential isoforms (DIs). Data represented as the total repeat bases normalized against 1 kb of total bases of the DTE/DIs. (E) Abundance of SINEs and LINEs family members within the promoter region of the DTEs and the DIs. (F) Binding sites of TFs overlapping with the repeat elements within the promoter region of the DTEs.

(DTE) followed their genome wide abundance at the promoter region. However, DIs differed significantly from genome-wide repeats distribution for SINEs, LINEs, and LTRs. MIR repeats were found to be highly abundant in the promoter regions of non-significant transcript isoforms (DI). We observed lower abundance of SINEs in the DTEs, while the abundance of LINEs and LTRs were higher in the DTEs compared to the DIs (Fig. 4D). However, abundance of Alu repeats was higher in the DTEs, albeit lower abundance of overall SINEs, suggesting a possible role of Alu but not the MIR in modulating the alternative splicing (Fig. 4E). We then predicted the binding sites of the TFs obtained from ChEA3, within the promoter region (5,000 bp upstream and 500 bp downstream of TSS) using Ciiider (Fig. 4F) (29). Consequently, many of these TFs had binding sites within the repeats as well, suggesting that repeat elements within the promoter region might play a role in expanding the repertoire of binding sites and thereby influencing the alternative splicing in COVID-19 patients (30). Together, these evidence possibly indicate that the genome-wide abundance of repeats was modulating the alternate transcription in the mortality patients, which when deviated in the DIs, led to insufficient alternative transcription (31). In other words, the differential repeats distribution between the DTE and DI in the mortality patients is important for the COVID-19 disease severity sub-phenotypes and the clinical outcome of the SARS-CoV-2 infected patients. This evidence merits a focused attention to understand the mechanism in future not only in COVID-19 but also in other RNA virus infections.

## Differential abundance of repeat elements within differentially expressed transcripts

Since we found repeat elements to be significantly associated with the promoter site of the transcripts, we looked within the transcripts as well for repeat element distribution and its possible role in alternative splicing (Fig. 4A). Besides, multiple studies have also reported repeat elements to be involved in regulation of alternative splicing (32, 33). We observed higher abundance of LINEs, LTRs, simple repeats, and DNA elements in the protein coding DTEs compared to the DIs (Fig. 5A; Table S6). Within the differentially expressed lncRNA transcripts, abundance of LINEs and DNA elements were significantly low within the DTEs compared to the DIs (*P* value < 0.001), while abundance of LTRs and simple repeats followed a trend like the protein coding transcripts (*P* value < 0.001) (Fig. 5B; Table S6). SINEs, LTRs, and simple repeat abundance were higher in the retained introns of the DTEs compared to the DIs, while abundance of LINEs and simple repeats were higher and LTRs, DNA elements and SINEs were lower in all other DTEs (Fig. 5C and D; Table S6). Notably, the abundance of the Alu elements was significantly lower in the DTEs (*P* value 0.012) (Fig. 5E to H). Except for the LINE1 abundance within the DTE lncRNAs, we observed an overall higher abundance of LINEs within the DTEs compared to the DIs (Fig. 5E to H). Based on the differential abundance of repeats within the DTEs and other studies reporting the role of repeats in the regulation of alternative splicing, we propose a possible repeat element-mediated regulation of alternative splicing of the DTEs in the COVID-19 mortality patients. But it merits focused study to understand the mechanism.

## Alu elements adjacent to splice sites are possible regulator of alternative splicing

To understand the role of repeat elements in regulating alternative splicing, we also looked at the repeat element distribution within the splicing sites of the 10 DTEs and

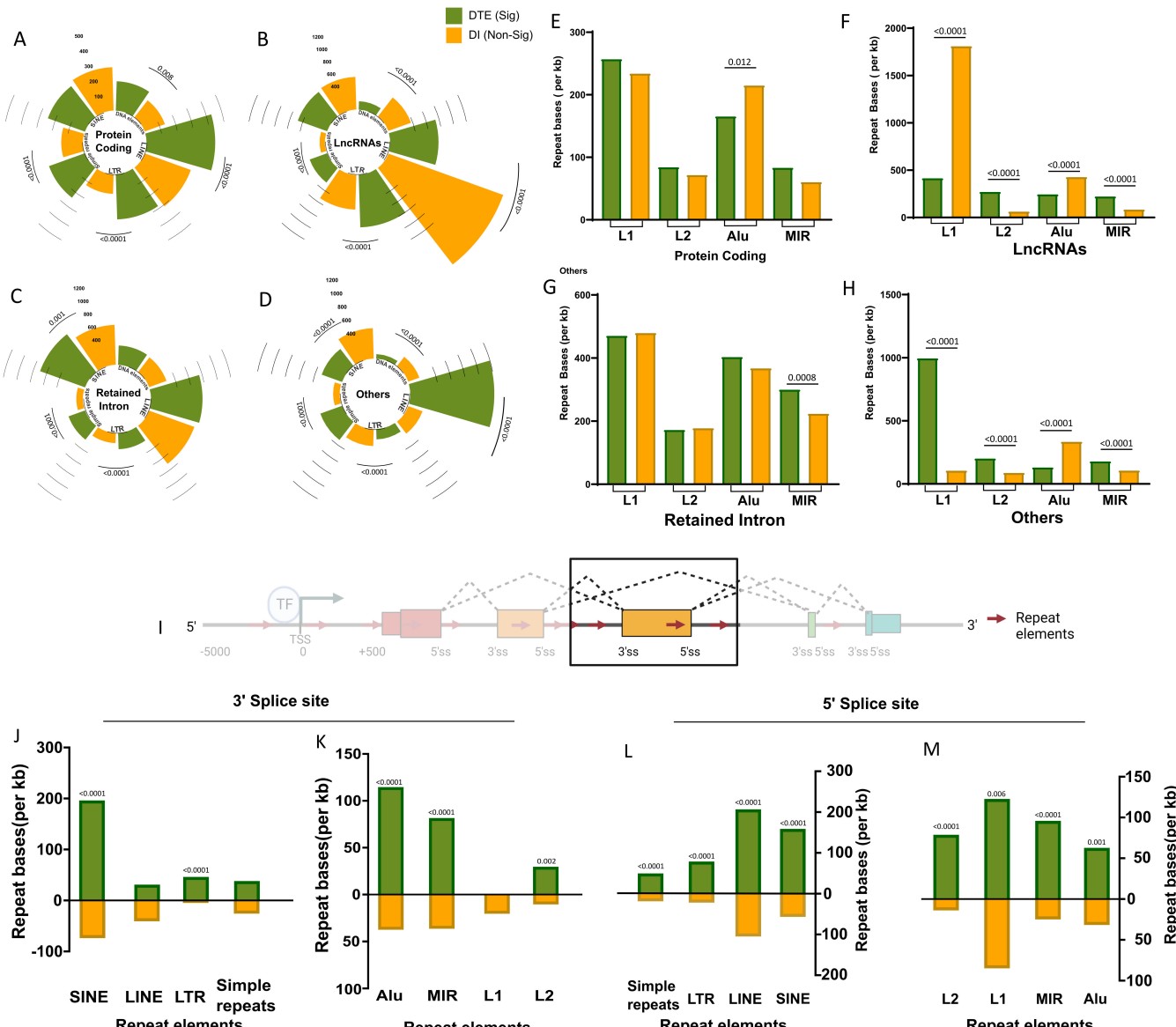

**FIG 5** Repeat element distribution within the differentially expressed transcripts and splice sites. Comparison of repeat element distribution within the coding regions of the differentially expressed transcript (DTE) and the differential isoforms (DI) of (A) protein coding genes, (B) LncRNAs, (C) retained introns, and (D) other biotypes. Distribution of repeat sub-families between the DTE and the DI of (E) protein coding genes, (F) LncRNAs, (G) retained introns, and (H) other biotypes. (I) Graphical representation highlighting the distribution of repeats near the 3′ and 5′ splice sites. (J and K) Distribution of repeat elements and sub-families at the 500 bp upstream of 3′ splice sites. (L and M) Distribution of repeat elements and sub-families at the 500 bp downstream of the 5′ splice sites. LINE, long interspersed nuclear element; LTR, long terminal repeat; SINE, short interspersed nuclear element; MIR, mammalian-wide interspersed repeat.

their non-significant DIs (Table S7). We selected regions of 500 bp upstream from the 3′ splice sites (3′ss) and downstream from the 5′ splice sites (5′ss) and scanned for the presence of repeat elements (Fig. 5I). Importantly, we observed a significantly higher abundance of SINEs (*P* value < 0.0001) and LTRs (*P* value < 0.0001) adjacent to the 3′ splice site (3′ss) or the upstream regions of the DTEs compared to the DIs (Fig. 5J; Table S7). Notably, abundance of both Alu and MIR elements were higher, while only LINE2 was significantly high near the 3′ss of DTEs (*P* values < 0.001) (Fig. 5K). The abundance of SINEs, LINEs, LTRs, and simple repeats as well as their family members were higher near the downstream splice site or the 5′ splice site (5′ss) within the DTEs compared to the DIs (Fig. 5L and M; Table S8).

Combining the repeat element abundance at 3′ss and 5′ss, we observe a significant presence of SINEs within the 500 bp upstream and downstream of the splice sites. Previous studies on the role of Alu in alternative splicing have reported intronic Alu to regulate alternative splicing when present near the splice sites (34). Therefore, we looked at the Alu sub-families and their abundance adjacent to the splice sites within the DTEs and the DIs. We discovered that Alu Y was present only near the 3′ss of DTEs, while abundance of Alu J was higher compared to the DIs (Fig. S2G). On the other hand, Alu Y was completely absent adjacent to the 5′ss of both the DTEs and the DIs, while Alu J was only present adjacent to the 5′ss of DTEs (Fig. S2H). Alu Y is the youngest member of the Alu family and is more actively involved in gene regulation, while Alu J is the oldest member of the Alu family. Therefore, higher abundance of Alu repeats, especially, Alu Y in the 3′ss within DTEs suggest plausible Alu-mediated regulation of alternative splicing within the COVID-19 mortality patients.

## DISCUSSION

Numerous studies have investigated the differential host transcriptomic response in the COVID-19 patients, however most of them have focused on the healthy vs disease comparisons at the gene level, with only a few focusing on the disease sub-pheno-types (18, 35, 36). In a disease like COVID-19, which has extremely heterogeneous clinical outcomes, it is especially important to investigate the causes for variable clinical outcomes (mild, moderate, severe, and mortality) in patients infected with the similar/same primary infectious agent—SARS-CoV-2. With more than 85% of the human genes expressing more than one transcript, it is also crucial to comprehend the transcript-level functional diversity between the disease sub-phenotypes. There-fore, in this first-of-its kind study, we explored the transcript diversity, followed by characterization of the differentially expressed transcripts within the COVID-19 severity sub-phenotypes. Furthermore, we propose possible transcription factor and repeat element mediated modulation of alternative transcription in the differentially expressed transcripts (Fig. 6).

As age can modulate disease severity, we compared it between the patient sub-groups. While the median age varied between mild and moderate/severe/mortality patients, there was no significant differences between other severity groups, *viz.* moderate vs severe, severe vs mortality, and moderate vs mortality. Thus, we checked the effect of age on transcriptome between mild and mortality; however, we found no significant association between the age and the significantly expressed transcripts. This suggests that despite age being an important component, it is not a major confounder alone. But in conjunction with comorbidities, treatment regimen, host immune response, and the viral strain, it may affect the expression of transcripts. Next, we compared the viral clade between different severity/outcomes and observed that despite different clinical severity, the underlying viral clade was similar (19A, 20A, and 20B) (Fig. 1F). We observed a lower Ct value in the group with higher severity compared to a milder patients (e.g., mortality compared to moderate). We also observed a similar trend in the SARS-CoV-2 viral reads from the RNA-seq data, where higher viral reads were detected in the moderate and severe patients compared to the mild patients. However, the correlation should be made with caution, mainly because several non-canonical factors (such as genomic rearrangements, trans-splicing, or transcriptional slippage) can affect the abundance of viral reads in a traditional RNA-seq data and more specialized approaches such as virus inclusive RNA-seq or dual RNA-seq are better suitable for this purpose (37, 38).

On comparison of the genes with differentially expressed transcripts to our previous study on COVID-19 host response at the gene level, it was revealed that most of the genes with transcript variants were not differentially expressed (18). This affirms the difference as well as granularity between the gene- and transcript-level expression patterns. We observed a decreased transcripts diversity in the mortality patients compared to the mild, moderate, and severe. However, we found significant

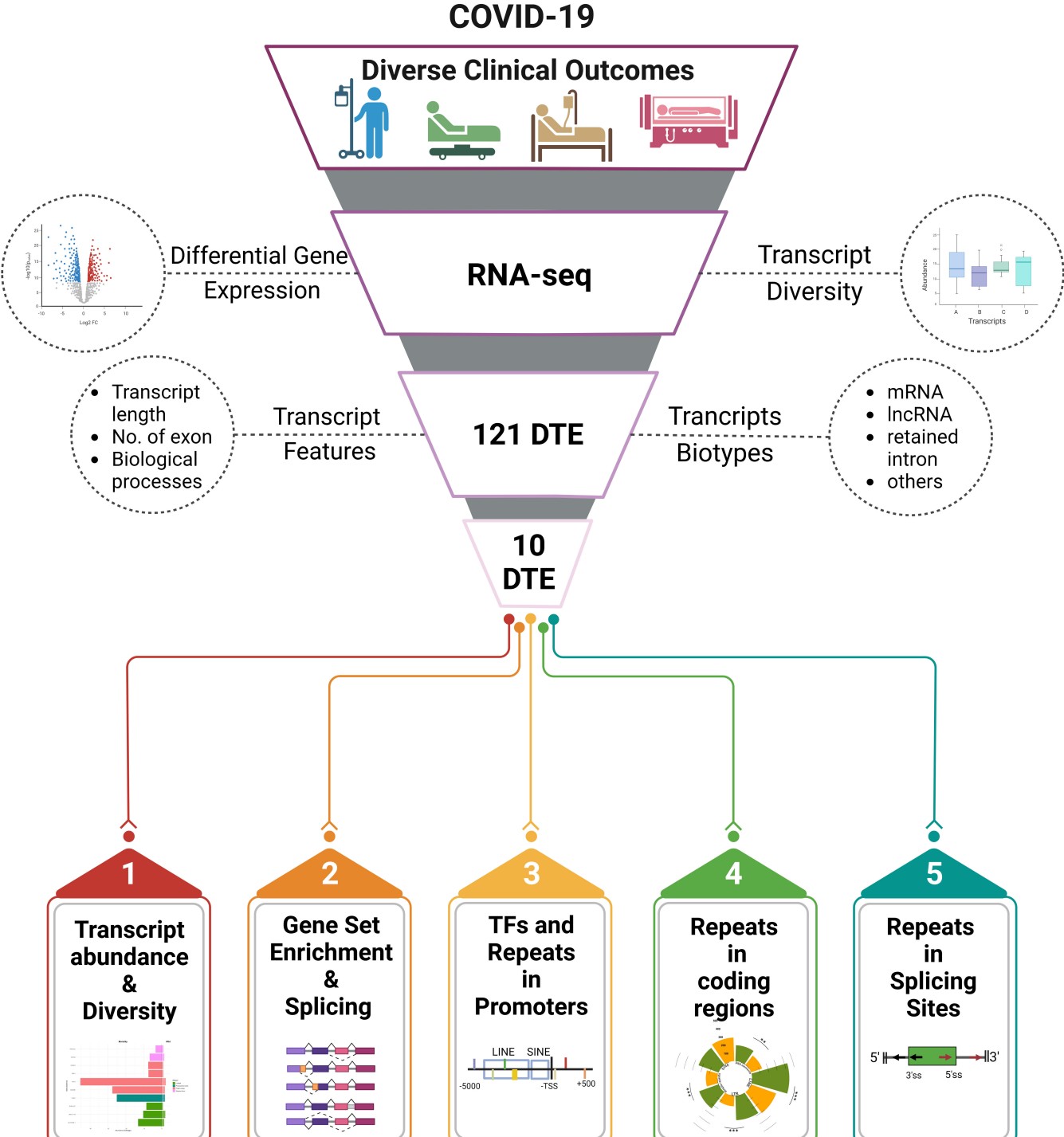

**FIG 6**  Summary of the study threading together the hierarchical inferences for the role of the alternate transcription in COVID-19. Through a series of interconnected analysis, the study highlights the role of alternative transcription in COVID-19 mortality cases and possible mechanisms of the alternative transcription.

transcript-level differential expression between the mild and the mortality patients only. The transcripts that were differentially expressed between mild and mortality were also differentially expressed in moderate/severe vs mortality, even though the difference in transcript level expression between the latter comparison groups was not statistically significant. This is most likely due to the moderate and severe groups' severity indices

being closer to those of the mortality. At the same time, since mild and mortality patients are quite contrasting with respect to the severity indices, the transcript isoform expression varied significantly with possible functional role. However, similar trends in both transcript diversity and transcript specific expression in moderate/severe vs mortality and mild vs mortality emphasize the significance of transcript-level understanding of the COVID-19 sub-phenotypes.

We observed an overall decreased transcript diversity and shorter transcript length with a smaller number of exons per transcripts in the mortality patients. Since there was no global transcriptional suppression, and absence of differential cell type abundance, we propose the decreased transcript diversity, abundance of shorter transcripts, and a smaller number of exons to be associated with the COVID-19 mortality. The selection toward combination of experimental as well as analytical approaches rule out the possible effects on the inferences drawn by the partly degraded RNA samples. Multiple studies have highlighted the dysregulation of transcript diversity during infection with the expression of shorter length transcripts as one of the consequences (11, 39, 40). The extreme diversity of putative 5′ss makes the process of recognition and selection of the 5′ss complex (41). The shorter isoform does not necessarily correlate with a smaller number of exons or general decrease in the transcription. Shorter isoforms are possible due to the expression of transcripts with shorter 3′UTR region, e.g., as seen during *M. tuberculosis* infection, as reported by Kalam et al. (12). On the other hand, not all infectious disease leads to selective expression of shorter transcripts, as reported in the case of *Anaplasma phagocytophilum* infection by Dumler et al. (42). Therefore, the shorter transcripts observed in our study are potentially specific to the SARS-CoV-2 infection and/or severity sub-phenotypes, rather than being associated with global transcription suppression or a smaller number of exons.

Therefore, we hypothesize that when the host is challenged with a severe infection, the cell probably starts generating shorter transcripts while bypassing the process of multiple splicing and joining of introns and exons to respond quickly. Another possible hypothesis is that the virus hijacks the host transcriptional machinery for its replication, which might lead to suppression of transcript diversity in case of severe infection (43, 44). Through the pathway enrichment as well as gene set enrichment analysis of the differentially expressed transcripts, we do observe a concerted effort by the virus to facilitate its entry, transportation, and subversion of immune response in the mortality patients. Interestingly, these pathways differ from those captured in our conventional differential gene expression analysis which highlighted dysregulation of innate immunity-related genes (18).

It is important to note that two significantly differentially expressed transcripts, *SRRM4* and *NAV3,* are mainly expressed in the nervous system. However, recent studies have reported association in some other diseases as well. For example, *SRRM4* and *NAV3* are reported to regulate lung cancer and colorectal cancer, respectively (24, 45). The European Molecular Biology Laboratory (EMBL) expression atlas shows their expression in different tissues, such as non-endothelial lung cells as well as other tissues in the lungs and gastrointestinal tract, in addition to the neuronal tissues. Most recently, both *SRRM4* and *NAV3* were reported to be upregulated in the COVID-19 patients' nasopharyngeal tissue, indicating yet to be discovered specific functional roles, especially during infectious diseases, and awaits investigating the functional role in a focused way.

To understand the splicing mechanism in the differentially expressed transcripts, we examined the different forms of alternative splicing. Generally, exon skipping is the most common splicing mechanism utilized; however, our results highlight the usage of alternative promoter site and alternative termination site as the two key splicing mechanisms in the DTEs. Recent studies have reported hijacking of the host splicing machinery by SARS-CoV-2 for its replication (11, 39). While the switch in the splicing mechanism might be attributed to the hijacking of the splicing machinery, this also supports our hypothesis of making shorter transcripts in quicker time during severe infection. Since alternative promoter site usage was also an important splicing

mechanism observed, we investigated the transcription factor binding site (TFBS) and repeat element abundance at the promoter region, since both binding of TFs and presence of repeat element are known to modulate the alternate splicing (32, 46). Alternative splicing is modulated by the speed of RNA Polymerase II mediated elongation (47, 48). Several TFs, including some of the TFs from our study (MECOM, HOXB13, FOSL1, and CTCF), are known to regulate the RNA polymerase II activity and elongation speed, thereby possibly affect alternative splicing. Moreover, significant high abundance of Alu repeats within the promoter region of the DTEs like genomic repeat distribution, and presence of binding sites of TFs within the repeats suggest a possible repeat element mediated regulation of alternative splicing by modulating the TF binding at the promoter region.

Repeat elements are reported to regulate alternative splicing not only at the promoter region but also at the splicing sites (34). Although we observed a differential abundance of repeat elements within the exonic region across the transcript biotypes (exonic region and part of the introns in retained introns) between the DTE and DI, the repeat elements abundance did not follow any pattern and seemed to be a genomic feature associated with the transcript biotype. However, the repeat element abundance near the splice site or the intronic regions adjacent to the splice site revealed a consistent higher abundance of SINE and LTRs at both ends of the splice site within the DTEs (Fig. 7). Both Alu and MIR elements of the SINE family are known to be GC rich (49, 50). Multiple studies have reported that higher GC content near the splice site facilitates alternative splicing by forming secondary structure near the splice site (51, 52). We also observed higher abundance of younger members of the Alu family, Alu Y, near the 3′ss, suggesting a more active role of Alu elements at the 3′ss in modulating the alternative splicing. Also, the presence of Alu Y adjacent to 3′ss means the right arm of the Alu, which is known to be actively involved in alternative splicing, to be in the proximity of the splice site. This, along with the fact that Alu Y is most active in terms of gene expression regulation among the Alu family members and our previous study suggesting the association of Alu elements with COVID-19 severity (36) led us to propose Alu-mediated alternative splicing as one of the possible modulators of disease severity. Although the role of LTRs in modulating alternative splicing is yet to be determined, its significant abundance at the splice site warrants further investigation. Overall, the results captured through the evidence from the data suggested that the intronic Alu near the splicing site, but not the exonic Alu, seemingly regulates alternative splicing underlying the mild and mortality COVID-19 disease outcomes. One of the major limitations of the study is the lack of transcript-specific pathway information for understanding transcript-specific functions. Because different transcript isoforms exhibit different expression patterns, understanding the specific function of the variably spliced transcripts is crucial. Furthermore, the study is based on samples gathered from patients on the day they were admitted to the hospital. Although the samples are optimal for examining the early host response to COVID-19, a longitudinal data can help elucidate the dynamics of alternative splicing during infection.

## Conclusion

In summary, we compared the transcriptomic signatures of 125 RT-PCR positive and sequencing confirmed SARS-CoV-2 hospital admitted patients with varying COVID-19 disease outcomes, ranging from mild, moderate, and severe to fatality. We performed RNA-seq using the nasopharyngeal swabs of the patients collected at the time of hospitalization. This was followed by transcript-level differential expression analysis which revealed evidence of decreased transcriptional diversity in the mortality patients when compared to the recovered with diverse clinical sub-phenotypes (mild, moderate, and severe). In the mortality patients, we found that promoter usage significantly differed between the differentially expressed transcripts and their non-significant isoforms. However, alternate termination precedes exon skipping events in the significantly expressed protein-coding transcripts. Finally, we also found differential

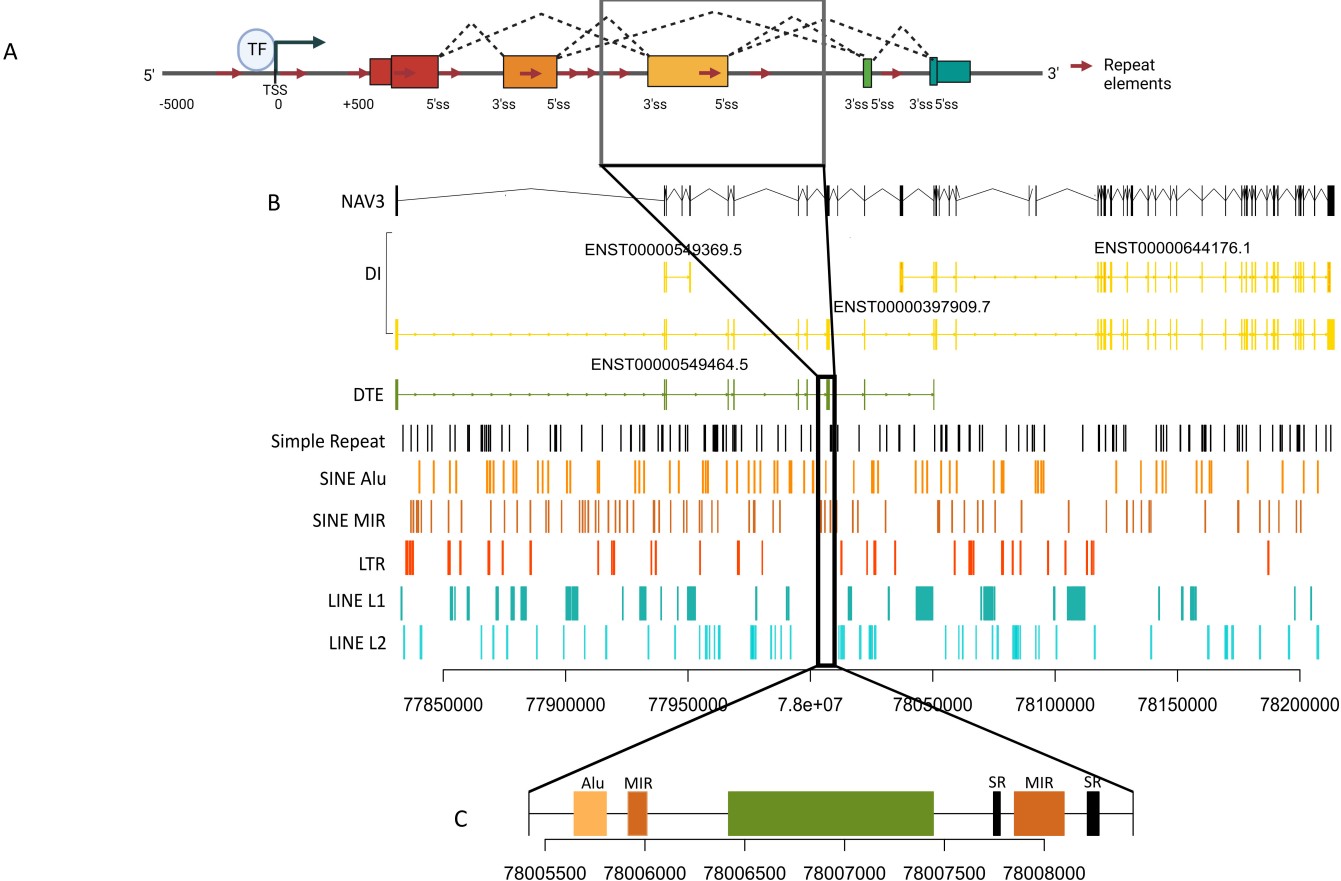

**FIG 7** Summary and possible repeat element-mediated alternative splicing in protein coding gene taking *NAV3* as an example case. (A) Graphical representation of possible repeat element mediated alternative splicing. (B) Genomic representation of *NAV3* gene (gray representing combined transcript, yellow representing DI, and green representing DTE) as well as distribution of repeat elements such as simple repeats, SINEs (Alu and MIR), LTR, and LINEs (L1 and L2) across the gene. (C) A zoomed-in view of repeat element distribution adjacent to the 3′ and 5′ss.

enrichment of repeat elements at the promoter region and specific enrichment of SINE (Alu) at the splicing sites of the significant transcripts, underscoring the plausible role of repeat mediated alternative splicing as a possible modulator of COVID-19 disease outcome.

The transcript-level differential expression analysis provides evidence for suppressed transcript diversity, length, and exons per transcripts in the COVID-19 mortality patients, possibly because of the host's attempt to rapidly mount a counter response under severe infection. Furthermore, the alternative splicing events in the COVID-19 mortality is modulated by switching towards non-canonical splicing mechanism and higher abundance of repeat elements at the promoter region and splicing sites. Thereby, in this study, we highlight the significance of alternative splicing in modulating COVID-19 disease severity sub-phenotypes and its possible mechanism.

## MATERIALS AND METHODS

### Sample collection and pre-processing, and diagnosis

The study was conducted on 125 patients admitted to a tertiary healthcare center in Delhi, India (MAX Superspeciality Hospital), with confirmed COVID-19 positive status based on qRT-PCR. Detailed clinical presentation, demographic data along with qRT-PCR results, and disease outcomes for each patient were retrieved from electronic medical

records and carefully documented for its usage during analysis. The nasopharyngeal swabs were collected in VTM by the hospital paramedical staff at the time of admission. Viral RNA from VTM solutions was isolated using QIAmp Viral Mini Kit (Qiagen, Cat. No. 52906) and SARS-CoV-2 detection and quantification was performed using TRUPCR SARS-CoV-2 Kit (3B BlackBio Biotech India Ltd., Cat. No. 3B304) with a cycle threshold of 35. These samples were further confirmed by whole genome sequencing of the SARS-CoV-2.

## Clinical sub-grouping based on clinical outcomes

The COVID-19 patients were initially categorized based on outcomes: recovered ($n =$ 107) and mortality ($n = 18$). The recovered individuals were further segregated into three severity sub-phenotypes: mild ($n = 62$), moderate ($n = 31$), and severe ($n = 14$) as per Indian Council of Medical Research guidelines. The clinical parameters taken into consideration were $SpO_2$ levels, requirement of respiratory support, and/or breathlessness. Patients with the $SpO_2$ level of ≥94% and no breathing problem were grouped as mild. Patients with breathing difficulty and $SpO_2$ levels ranging between 91%–93% were categorized as moderate. Patients with respiratory distress, $SpO_2$ levels <90%, and requiring respiratory support were classified as severe. The patients who succumbed to COVID-19 during hospital stay are grouped as mortality.

## Library preparation and sequencing

Sequencing libraries were prepared from 250 ng of nasopharyngeal RNA (collected on the day of admission) using Illumina TruSeq Stranded Total RNA Library Prep Gold (Cat. No. 20020599) as per manufacturer's reference guide (1000000040499 v00) and our previous study (53). Briefly, target-specific biotinylated oligos with Ribo-Zero rRNA removal beads were used to remove cytoplasmic and mitochondrial rRNA followed by fragmentation of purified RNA using divalent cation under elevated temperature. The first-strand cDNA was synthesized from the fragmented RNA using random primers and SuperScript IV reverse transcriptase. The second-strand cDNA was synthesized using DNA polymerase 1, post degradation of the RNA strand from the previous step using RNaseH. The 3′ blunt end of the double-stranded cDNA was adenylated, followed by indexing and amplification. The final library was purified using AMPure XP beads, with beads to sample ratio of 1:1 (Beckman Coulter, A63881). Agilent 2100 bioanalyzer was used to check the library quality, followed by subsequent denaturation using 0.2 N NaOH and sequencing on NovaSeq 6000 using NovaSeq S2 v1.5 reagents at $2 \times 101$ read length and loading concentration of 450 pM.

## Differential transcript expression analysis

The raw sequencing reads were quality checked using FastQC and trimmed with Trimmomatic v.0.39 to remove low quality bases (54). The reads were then re-assessed with FastQC to confirm quality improvements (Fig. 8).

The filtered reads were then quantified against the Human reference genome (GRCh38.106 primary assembly from Ensembl) using Salmon (v.1.8.0) (55). A differential gene expression analysis was performed from the Salmon quantified reads using DESeq2 package (56). In Salmon, --numGibbsSamples parameter with 20 inferential replicates was used to generate bootstrap abundance estimates for each sample using posterior Gibbs sampling. The quantification files generated by Salmon were imported to R environment using tximport for differential transcript-level expression analysis of RNA-seq using inferential replicate counts with swish method in Bioconductor package fishpond (v.2.0.1) (57). The swish method extends on SAMseq was implemented in the samr package by considering inferential uncertainty and allowing control for batch effects. Differential transcript-level analysis was performed for mortality vs mild/moderate/severe. The Benjamini-Hochberg correction was used to correct for multiple comparisons (with an FDR cut-off <0.05). Differentially expressed transcripts with

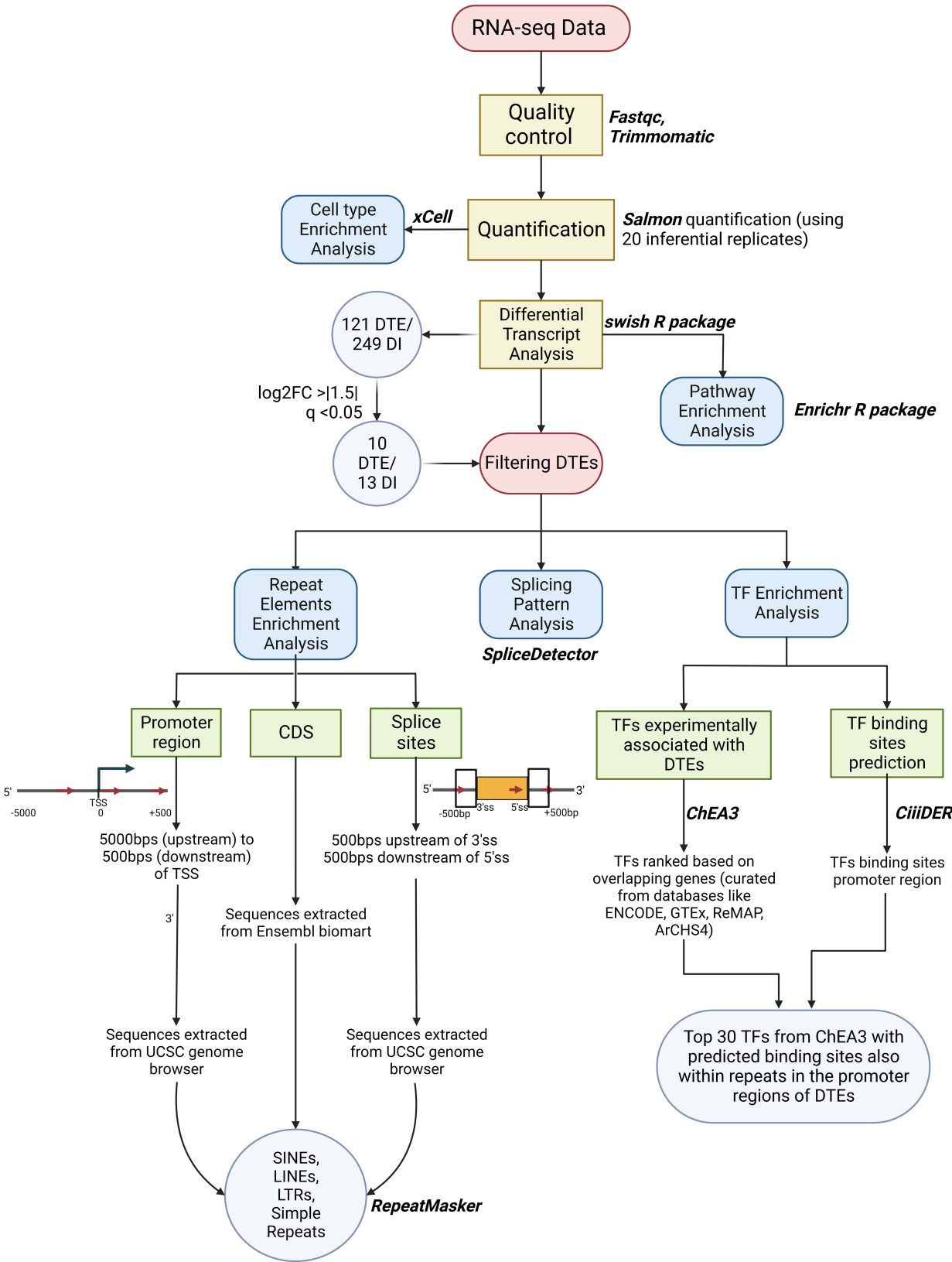

**FIG 8** Flowchart of the steps followed for differential transcript level analysis of the RNA-seq samples. CDS, coding sequence; DI, differential isoform; DTE, differentially expressed transcript; TF, transcription factor.

q-value < 0.05 were considered as significant and $\log_2$ fold change of >1.5 was further applied to make the selection more stringent. For the SARS-CoV-2 gene expression, fastq files were aligned to SARS-CoV-2 reference genome (MN908947.3) using STAR aligner. Reads were then counted using HTSeq gene count (58). Differential expression analysis was performed using the DESeq2 package. Genes with log2FC > |1.5| and $P$-adj < 0.05 were considered significant.

## Cell type enrichment analysis

The xCell R package was used to perform the cell type enrichment study (59). It uses gene expression data to perform cell type enrichment analysis using the 64 immunological and stromal cell types information. It is based on gene signatures learned from thousands of pure cell types from various sources. The Mann-Whitney $U$-test was used to compare cell types between the group of patients.

## Pathway enrichment analysis

Function enrichment of DTEs was performed using the Enrichr package against the KEGG database (60). Pathways with statistically significant $P$-value cutoff <0.05 were considered. The pathways were plotted using the ggplot2 R package against the combined score and number of genes involved in the pathways. Redundant pathways were excluded. We also performed gene set enrichment analysis (GSEA) against reactome database using fgsea R package. We selected a $P$-value cutoff of < 0.1 (61).

## Repeat elements enrichment analysis

We looked at the repeat elements at three different genomic regions: the coding sequence (CDS), the promoter region, and the splice sites for the DTEs and their non-significantly expressed isoforms (referred to as differential isoforms ). The CDS sequences were extracted from the Emsembl bioMart. For the promoter region analysis, we took 5,000 bases upstream and 500 bases downstream from the transcription start site for the DTEs, DIs, and all the protein-coding and non-protein coding transcripts in the human genome to determine a null distribution of repeats in the promoter region. The genomic loci for the TSS site for each transcript was extracted from the UCSC Genome Browser and the promoter sequences were obtained from the Ensembl bioMart. Finally, for the splice sites, we selected a window of 500 bases upstream from the 5′ss and 500 bases. A total of 131,038 transcripts promoter regions were taken into consideration to determine the global null distribution of the repeats upstream from the 3′ss. To query for the repeat elements, all the sequences were uploaded to RepeatMasker using hmmer search engine and Human as DNA source (62). The repeat elements considered were, long interspersed nuclear elements (LINEs) (L1 and L2), short interspersed nuclear elements (SINEs) (Alu and MIR), long terminal repeat (LTR), DNA elements, and the simple repeats. The repeat bases were normalized for the transcript length and has been represented as repeat bases per kilobase. The significance between the DTE and the DIs were calculated using $\chi^2$ test.

## Transcription factor enrichment analysis

The enrichment of previously annotated TFs known to regulate the DTE by binding near the coding regions were obtained using the ChEA3 web-based tool (28). ChEA3 builds upon ChIP-seq and RNA-seq data extracted from multiple sources [ENCODE (The Encyclopedia of DNA Elements), ReMap (database of transcriptional regulators peaks derived from curated ChIP-seq, ChIP-exo, DAP-seq experiments in the Human, Mouse, Fruit Fly, and Arabidopsis Thaliana), GTEx (Genotype Tissue Expression provides study tissue-specific gene expression and regulation studies), and ARCHs4 (All RNA-seq and ChIP-seq sample and signature search provides gene and transcript counts from the RNA-seq experiments in GEO and SRA databases) and publications)] allowing for the integrative analysis. Based on the overlap between the given list of differentially

expressed genes, ChEA3 predicts TFs associated with user-input set of genes. We next used CiiiDer to predict the TFBS in the promoter region for each DTE (29). From the predicted TFBS, those within repeat elements and obtained from the ChEA3 tool, were filtered for functional inference.

## Detecting splicing pattern

The SpliceDetector (v.1.0.0.0) tool was used to detect alternative splicing patterns for protein coding genes (21). SpliceDetector employs transcript identifiers to detect the alternate splicing events using splice graphs. It provides the frequency of active splice sites in the pre-mRNA. The splicing pattern was detected between the significant transcripts and the non-significant transcript isoforms and the $\chi^2$ test was performed to check for significant differences between the different splicing patterns.

## Phylogenetic analysis

The FASTA sequences obtained from the whole genome sequencing of SARS-CoV-2 were used for phylogenetic analysis. Sequences with coverage <50% were filtered out. The remaining 84 sequences were aligned against the SARS-CoV-2 reference genome (MN908917.3) using MAFFT (63). The edges of the sequences were trimmed, and phylogenetic tree was built using FASTME algorithm in the NGphylogeny tool (64, 65). The newick was visualized in Figtree (http://tree.bio.ed.ac.uk/software/figtree/) and clade annotations were added from the Nextclade database (https://clades.nextstrain.org/).

## Statistical analysis and data visualization

Wherever appropriate, we compared the differences between the data points using the two-tailed Mann-Whitney U-test, and Chi-square testing. We performed simple linear regression to find association between the age and the 10 differentially expressed transcripts. The statistical tests were performed using a licensed version of GraphPad Prism. The ggbio (v.1.44.1), GenomicFeatures (v.1.48.3), ggplot2 (v.3.3.3), EnhancedVolcano (v.1.14.0), Gviz (v.1.40.1), tracklayer (v.1.56.1), and trackviewer (v.1.32.1) R packages were used for data visualization (66–70). The P value < 0.05 was considered as statistically significant unless stated otherwise.

## ACKNOWLEDGMENTS

The authors duly acknowledge all the COVID-19 patients who participated in the study. Authors acknowledge the help and support from Dr. Aradhita Baral and Dr. Bharti Kumari toward facilitation as Research manager and coordination with the funders. Authors acknowledge the support of Anil Kumar and Nisha Rawat toward COVID-19 sample transport and sample management. P.C. acknowledges the CSIR for his Research Fellowship.

P.M.: formal analysis, writing - original draft, and visualization. P.C.: investigation, formal analysis, writing - original draft. V.R.: formal analysis. R.P.: conceptualization, writing - review & editing, supervision and funding acquisition. All authors discussed the results and contributed to the final manuscript.

This research was funded by the Bill and Melinda Gates Foundation, grant number INV-033578.

The authors have declared that no conflict of interest exists. The funding body did not have any role in designing the study or the interpretation of the results.

## AUTHOR AFFILIATIONS

[1]Division of Immunology and Infectious Disease Biology, INtegrative GENomics of HOst-PathogEn (INGEN-HOPE) Laboratory, CSIR-Institute of Genomics and Integrative Biology (CSIR-IGIB), Delhi, India
[2]Academy of Scientific and Innovative Research (AcSIR), Ghaziabad, India

³Max Super Speciality Hospital (A Unit of Devki Devi Foundation), Max Healthcare, Delhi, India

## AUTHOR ORCIDs

Priyanka Mehta ⓘ http://orcid.org/0000-0001-6298-4322
Rajesh Pandey ⓘ http://orcid.org/0000-0002-4404-8327

## FUNDING

| Funder | Grant(s) | Author(s) |
| --- | --- | --- |
| Bill and Melinda Gates Foundation (GF) | INV-033578 | Rajesh Pandey |

## AUTHOR CONTRIBUTIONS

Priyanka Mehta, Data curation, Formal analysis, Investigation, Methodology, Visualization, Writing – original draft | Partha Chattopadhyay, Formal analysis, Investigation, Methodology, Visualization, Writing – original draft | Varsha Ravi, Formal analysis, Investigation, Visualization | Bansidhar Tarai, Resources | Sandeep Budhiraja, Resources | Rajesh Pandey, Conceptualization, Data curation, Funding acquisition, Investigation, Project administration, Resources, Supervision, Writing – review and editing

## DATA AVAILABILITY

All raw and processed sequencing data generated in this study have been submitted to the NCBI Sequence Reads Archive under BioProject accession number PRJNA678831. The RNAseq QC files are available in Zenodo at 7471319. The GISAID IDs of the consensus FASTA submitted to GISAID-EpiCoV are mentioned in Table S1 in the supplemental material.

## ETHICS APPROVAL

The studies involving human participants were reviewed and approved by CSIR-IGIB's Human Ethics Committee Clearance (Ref No: CSIR-IGIB/IHEC/2020-21/01). The patients/participants provided their written informed consent prior to participation in this study.

## ADDITIONAL FILES

The following material is available online.

### Supplemental Material

**Figure S1 (Spectrum01351-23-s0001.pdf).** Clinical features and transcript diversity characterization across moderate and severe patients.
**Figure S2 (Spectrum01351-23-s0002.pdf).** Splicing pattern and repeat element distribution in the differentially expressed transcripts.
**Table S1 (Spectrum01351-23-s0003.xls).** Clinical characteristics of the patients.
**Table S2 (Spectrum01351-23-s0004.xls).** Differential gene expression analysis of viral genes.
**Table S3 (Spectrum01351-23-s0005.xls).** Differentially expressed genes between mortality versus severe, mortality versus moderate, and mortality versus mild.
**Table S4 (Spectrum01351-23-s0006.txt).** Differentially expressed transcripts between mortality and mild groups.
**Table S5 (Spectrum01351-23-s0007.xls).** Promoter region repeat elements.
**Table S6 (Spectrum01351-23-s0008.xls).** Coding sequence repeat element distribution.
**Table S7 (Spectrum01351-23-s0009.xls).** Genomic coordinates of 10 DTEs and their non-significantly expressed DIs.
**Table S8 (Spectrum01351-23-s0010.xls).** 3' and 5' splice site repeat distribution.

Open Peer Review

**PEER REVIEW HISTORY (review-history.pdf).** An accounting of the reviewer comments and feedback.

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
