## [Reviewer comments · Microbiology Spectrum]

Microbiology Spectrum

SARS-CoV-2 infection Severity and Mortality is modulated by Repeat-mediated regulation of Alternative Splicing

Rajesh Pandey, Priyanka Mehta, Partha Chattopadhyay, Varsha Ravi, Bansidhar TARAI, and Sandeep Budhiraja

Corresponding Author(s): Rajesh Pandey, CSIR Institute of Genomics & Integrative Biology

Review Timeline:

Submission Date:	March 29, 2023
Editorial Decision:	July 6, 2023
Revision Received:	July 13, 2023
Editorial Decision:	July 14, 2023
Revision Received:	July 14, 2023
Accepted:	July 16, 2023

Editor: Yun Young Go

Reviewer(s): Disclosure of reviewer identity is with reference to reviewer comments included in decision letter(s). The following individuals involved in review of your submission have agreed to reveal their identity: Elif Yilmaz Gulec (Reviewer #4); Gaurav Verma (Reviewer #5)

Transaction Report:

DOI: <https://doi.org/10.1128/spectrum.01351-23>

July 6, 2023

Dr. Rajesh Pandey
CSIR Institute of Genomics & Integrative Biology
Integrative Genomics of Host Pathogen Laboratory
Mall Road
New Delhi, Delhi 110007
India

Re: Spectrum01351-23 (SARS-CoV-2 infection Severity and Mortality is modulated by Repeat-mediated regulation of Alternative Splicing)

Dear Dr. Rajesh Pandey:

Link Not Available

Sincerely,

Yun Young Go

Journals Department
Reviewer comments:

Reviewer #4 (Comments for the Author):

This is a well written manuscript, I enjoyed reading it.
I would like to comment about two points:

1) About the virus, did you make RT-PCR for the virus subtypes? Which SARS-CoV-2 variant was present in the patients? Did all the patents have the same virus variant? I think this is another criteria about the clinical outcome, some subtypes are more virulent. Could you please mention and discuss about it in the discussion or in the material methods section.

2) We know that COVID19 is more fatal in older age group as you mentioned in the text. Did you also compare the age groups for the differences in transcriptome. Are the changes related to mortality more abundant in the old patients? If possible you can compare the age groups and discuss about the effect of aging on the transcriptome, if there is no difference between the age and clinical severity groups.

Reviewer #5 (Public repository details (Required)):

If the manuscript is accepted the transcriptome data from sequencing of 125 hospital admitted COVID-19 patients needs to be submitted on GEO datasets.

Reviewer #5 (Comments for the Author):

The current study reveal a novel mechanism of alternative splicing in COVID-19 differential studies. The importance of transcriptomic analysis and possible mechanism of action is very well highlighted by this study. However, I have a few minor questions that you need to address.

1- Did you validate using another pathways tool whether you observe consistent finding?

2- What was the concentration of AMPure beads used to purify the library?

3- Figure 2 labelling and text is not matching. For eg. fig. 2 (D-H) are not correctly labelled according to the text, please correct it. Fig 3 (GH) graph boundaries thickness is inconsistent to Fig (IJ), keep it consistent.

4- What are the limitation of your study?

Staff Comments:

Preparing Revision Guidelines

Please return the manuscript within 60 days; if you cannot complete the modification within this time period, please contact me. If you do not wish to modify the manuscript and prefer to submit it to another journal, please notify me of your decision immediately so that the manuscript may be formally withdrawn from consideration by Microbiology Spectrum.

Dear Authors,

This is a well written manuscript, I enjoyed reading it.

I would like to comment about two points:

- 1) About the virus, did you make RT-PCR for the virus subtypes? Which SARS-CoV-2 variant was present in the patients? Did all the patients have the same virus variant? I think this is another criteria about the clinical outcome, some subtypes are more virulent. Could you please mention and discuss about it in the discussion or in the material methods section.
- 2) We know that COVID19 is more fatal in older age group as you mentioned in the text. Did you also compare the age groups for the differences in transcriptome. Are the changes related to mortality more abundant in the old patients? If possible you can compare the age groups and discuss about the effect of aging on the transcriptome, if there is no difference between the age and clinical severity groups.

Dear Editor and the Reviewers,

We would like to take this opportunity to thank you for your valuable time and effort towards providing a thorough assessment of our research article, as well as helpful ideas for enhancing the inferences presented in the manuscript.

During the revised manuscript submission, we have addressed all the suggestions as applicable with additional analysis, figures, and supplemental material.

Best wishes,

Rajesh

Reviewer #4 (Comments for the Author):

This is a well written manuscript, I enjoyed reading it.

Thank you for your appreciation of our effort towards putting together the findings together in this manuscript covering an unexplored aspect of host response during COVID-19.

I would like to comment about two points:

1) About the virus, did you make RT-PCR for the virus subtypes? Which SARS-CoV-2 variant was present in the patients? Did all the patients have the same virus variant? I think this is another criteria about the clinical outcome, some subtypes are more virulent. Could you please mention and discuss about it in the discussion or in the material methods section.

We thank the reviewer for this insightful perspective.

We would like to share that these patients were first tested for COVID-19 positivity using RT-PCR. Subsequently, these RT-PCR patients are undertaken for SARS-CoV-2 whole genome sequencing. Thereafter, the viral phylogeny and identification of the variant have been determined using the genome sequences obtained from the Oxford Nanopore Platform.

As these COVID-19 patients are from pre-VOC (variants of concern) time point, majority of them have similar variant which lead us to hypothesize that despite having a similar variant, why the patients experienced differential disease severity levels (mild, moderate, severe, mortality).

We have included the clade information in the **Supplementary table 1** as well as added a phylogenetic tree of the clades in the **Figure 1**. We have also added a paragraph in both the discussion and methods section as well about the patient SARS-CoV-2 clades in this study. It is also important to share that these SARS-CoV-2 sequences are uploaded to GISAID (IDs included in the (Supplementary table 1). We have added the following section to the result.

“We also sequenced the whole genome of the SARS-CoV-2 virus isolated from nasopharyngeal swabs of the patients to determine whether patients with varying severity levels are infected with different strains of the virus. Despite the differences in clinical severity and outcome, we discovered that the virus strain (19A, 20A and 20B) was similar between mild, moderate, severe and mortality patients (Figure 1F). Overall, these clinical, sequencing, and demographic data represent the diversity of symptoms within the COVID-19 sub-phenotypes despite similarity in the underlying viral infection and emphasize the need of understanding the transcriptional dynamics within the COVID-19 severity sub-phenotypes.”-Page no 6, line no. 137-142.

Figure 1: Phylogenetic tree of the SARS-CoV-2 clades from positive patients. Sample labelled with pink colour belong to 20B. Disease severity types and SARS-CoV-2 lineages are distributed across the phylogeny as represented by the color of nodes (green for mild, yellow for moderate, blue for severe, red for mortality).

2) We know that COVID19 is more fatal in older age group as you mentioned in the text. Did you also compare the age groups for the differences in transcriptome. Are the changes related to mortality more abundant in the old patients? If possible you can compare the age groups and discuss about the effect of aging on the transcriptome, if there is no difference between the age and clinical severity groups.

We thank the reviewer for this important observation.

While the patient grouping was done with respect to the severity parameter as per ICMR (Indian Council of Medical Research) guidelines wherein SpO₂ levels and Ct values are important parameters, we observed that there was a significant age difference ONLY between the Mild and Moderate/Severe/Mortality groups as shown in the **Figure 2** below. However, there was NO significant difference between other severity groups, viz. moderate vs severe, severe vs mortality, moderate vs mortality.

However, the 10 differentially expressed transcripts (including one pseudogene, AL731559.1) have an average abundance greater in the Mortality group patients falling above the median age of 61 compared to the ones below that age group **Figure 3**. To check for its possible modulation due to age, we also performed the simple linear regression analysis between the age and expressions of the 10 differentially expressed transcripts between Mortality and Mild patients.

Importantly, we do not see a significant association between the transcripts and the age of the patients. This suggests that despite age being an important component, it is not a major confounder alone. But in conjunction with comorbidities, treatment regimen, host immune response and the viral strain, it may affect the expression of these transcripts. We have added a paragraph of the same in the discussion of the revised manuscript as below:

“As age can modulate disease severity, we compared it between the patient sub-groups. While the median age varied between mild and moderate/severe/mortality patients, there was no significant differences between other severity groups, viz. moderate vs severe, severe vs mortality, and moderate vs mortality. Thus, we checked the effect of age on transcriptome between mild and mortality, however, found no significant association between age and the significantly expressed transcripts. This suggests that despite age being an important component, it is not a major confounder alone. But in conjunction with comorbidities, treatment regimen, host immune response and the viral strain, it may affect the expression of transcripts. Next, we compared the viral clade between different severity/outcomes and

observed that despite different clinical severity, the underlying viral clade was similar (19A, 20A, 20B) (Figure 1F).” -Page no. 16, line no. 358-368.

Figure 2: Age distribution between mild, moderate, severe and mortality patients.

Figure 3: Average abundance distribution of mortality patients divided based on their median age. Patients above 61 years of age and patients below 61 years of age. It does not include the pseudogene, AL731559.1.

Reviewer #5 (Public repository details (Required)):

If the manuscript is accepted the transcriptome data from sequencing of 125 hospital admitted COVID-19 patients needs to be submitted on GEO datasets.

We thank the reviewer for this suggestion.

The raw sequencing fastq files as well as the metadata are uploaded and available at the NCBI Sequence Reads Archive (SRA; <https://www.ncbi.nlm.nih.gov/sra/>) under BioProject accession number **PRJNA678831**. The RNAseq QC files are available at (<https://doi.org/10.5281/zenodo.7471319>).

Reviewer #5 (Comments for the Author):

The current study reveal a novel mechanism of alternative splicing in COVID-19 differential studies. The importance of transcriptomic analysis and possible mechanism of action is very well highlighted by this study. However, I have a few minor questions that you need to address.

We thank the reviewer for appreciating the findings presented within the manuscript elucidating an unexplored aspect of COVID-19 for granular understanding of the COVID-19 differential disease severity types.

1- Did you validate using another pathways tool whether you observe consistent finding?

We thank the reviewer for this suggestion.

Based on your suggestion, we have used reactome pathways and performed gene set enrichment analysis using fgsea package in R. Taking a cutoff of $p \text{ value} < 0.1$, we find similar pathways such as integrin cell surface interactions pathway, extracellular matrix organisation pathways to be negatively enriched in the mortality patients which is consistent with the pathways obtained by the KEGG database, where we find suppression of cell-cell adhesion regulation as well as integrin-mediated signalling pathways as shown in **Figure 4** below .

Figure 4: Pathway enrichment analysis using reactome for 121 differentially expressed transcripts.

2- What was the concentration of AMPure beads used to purify the library?

We thank the reviewer for this query. The library was purified using AMPure XP beads, with bead to sample ratio of 1:1.

3- Figure 2 labelling and text is not matching. For eg. fig. 2 (D-H) are not correctly labelled according to the text, please correct it. Fig 3 (GH) graph boundaries thickness is inconsistent to Fig (IJ), keep it consistent.

We thank the reviewer for bringing it to our notice. During the revised submission, we have now corrected the Figure legend 2 as well as made the graph thickness uniform in Figure 3 (GH).

4- What are the limitation of your study?

We acknowledge the reviewer suggestion. In the updated submission, we have added a section discussing the study's limitations. The section is as follows:

“One of the major limitations of the study is the lack of transcript-specific pathway information for understanding transcript-specific functions. Because different transcript isoforms exhibit different expression patterns, understanding the specific function of the variably spliced transcripts is crucial. Furthermore, the study is based on samples gathered from patients on the day they were admitted to the hospital. Although the samples are optimal for examining the early host response to COVID-19, a longitudinal data can help elucidate the dynamics of alternative splicing during infection.” -Page 20 lines 475-482.

July 14, 2023

Dr. Rajesh Pandey
CSIR Institute of Genomics & Integrative Biology
Integrative Genomics of Host Pathogen Laboratory
Mall Road
New Delhi, Delhi 110007
India

Re: Spectrum01351-23R1 (SARS-CoV-2 infection Severity and Mortality is modulated by Repeat-mediated regulation of Alternative Splicing)

Dear Dr. Rajesh Pandey:

1. Some of the answers to the reviewers' comments were only provided in the rebuttal letter (e.g., Reviewer #5's comments 1 and 2). Please consider incorporating them in the main text accordingly.
2. Please check typographical, grammatical and formatting errors throughout the manuscript before the acceptance.

Thank you for submitting your manuscript to Microbiology Spectrum. As you will see your paper is very close to acceptance. Please modify the manuscript along the lines I have recommended. As these revisions are quite minor, I expect that you should be able to turn in the revised paper in less than 30 days, if not sooner. If your manuscript was reviewed, you will find the reviewers' comments below.

When submitting the revised version of your paper, please provide (1) point-by-point responses to the issues raised by the reviewers as file type "Response to Reviewers," not in your cover letter, and (2) a PDF file that indicates the changes from the original submission (by highlighting or underlining the changes) as file type "Marked Up Manuscript - For Review Only". Please use this link to submit your revised manuscript. Detailed instructions on submitting your revised paper are below.

Link Not Available

Sincerely,

Yun Young Go

Reviewer comments:

Preparing Revision Guidelines

- Point-by-point responses to the issues raised by the reviewers in a file named "Response to Reviewers," NOT IN YOUR COVER LETTER.
- Upload a compare copy of the manuscript (without figures) as a "Marked-Up Manuscript" file.
- Each figure must be uploaded as a separate file, and any multipanel figures must be assembled into one file.

- Manuscript: A .DOC version of the revised manuscript
- Figures: Editable, high-resolution, individual figure files are required at revision, TIFF or EPS files are preferred

Please return the manuscript within 60 days; if you cannot complete the modification within this time period, please contact me. If you do not wish to modify the manuscript and prefer to submit it to another journal, please notify me of your decision immediately so that the manuscript may be formally withdrawn from consideration by Microbiology Spectrum.

Dear Editor and the Reviewers,

We would like to take this opportunity to thank you for your valuable time and effort towards providing a thorough assessment of our research article, as well as helpful ideas for enhancing the inferences presented in the manuscript.

During the revised manuscript submission, we have addressed all the suggestions as applicable with additional analysis, figures, and supplemental material.

Best wishes,

Rajesh

Reviewer #4 (Comments for the Author):

This is a well written manuscript, I enjoyed reading it.

Thank you for your appreciation of our effort towards putting together the findings together in this manuscript covering an unexplored aspect of host response during COVID-19.

I would like to comment about two points:

1) About the virus, did you make RT-PCR for the virus subtypes? Which SARS-CoV-2 variant was present in the patients? Did all the patents have the same virus variant? I think this is another criteria about the clinical outcome, some subtypes are more virulent. Could you please mention and discuss about it in the discussion or in the material methods section.

We thank the reviewer for this insightful perspective.

We would like to share that these patients were first tested for COVID-19 positivity using RT-PCR. Subsequently, these RT-PCR patients are undertaken for SARS-CoV-2 whole genome sequencing. Thereafter, the viral phylogeny and identification of the variant have been determined using the genome sequences obtained from the Oxford Nanopore Platform.

As these COVID-19 patients are from pre-VOC (variants of concern) time point, majority of them have similar variant which lead us to hypothesize that despite having a similar variant, why the patients experienced differential disease severity levels (mild, moderate, severe, mortality).

We have included the clade information in the **Supplementary table 1** as well as added a phylogenetic tree of the clades in the **Figure 1**. We have also added a paragraph in both the discussion and methods section as well about the patient SARS-CoV-2 clades in this study. It is also important to share that these SARS-CoV-2 sequences are uploaded to GISAID (IDs included in the (Supplementary table 1). We have added the following section to the result.

“We also sequenced the whole genome of the SARS-CoV-2 virus isolated from nasopharyngeal swabs of the patients to determine whether patients with varying severity levels are infected with different strains of the virus. Despite the differences in clinical severity and outcome, we discovered that the virus strain (19A, 20A and 20B) was similar between mild, moderate, severe and mortality patients (Figure 1F). Overall, these clinical, sequencing, and demographic data represent the diversity of symptoms within the COVID-19 sub-phenotypes despite similarity in the underlying viral infection and emphasize the need of understanding the transcriptional dynamics within the COVID-19 severity sub-phenotypes.”-Page no 6, line no. 136-141.

Figure 1: Phylogenetic tree of the SARS-CoV-2 clades from positive patients. Sample labelled with pink colour belong to 20B. Disease severity types and SARS-CoV-2 lineages are distributed across the phylogeny as represented by the color of nodes (green for mild, yellow for moderate, blue for severe, red for mortality).

2) We know that COVID19 is more fatal in older age group as you mentioned in the text. Did you also compare the age groups for the differences in transcriptome. Are the changes related to mortality more abundant in the old patients? If possible you can compare the age groups and discuss about the effect of aging on the transcriptome, if there is no difference between the age and clinical severity groups.

We thank the reviewer for this important observation.

While the patient grouping was done with respect to the severity parameter as per ICMR (Indian Council of Medical Research) guidelines wherein SpO₂ levels and Ct values are important parameters, we observed that there was a significant age difference ONLY between the Mild and Moderate/Severe/Mortality groups as shown in the **Figure 2** below (included as **Supplementary figure S1A**). However, there was NO significant difference between other severity groups, viz. moderate vs severe, severe vs mortality, moderate vs mortality.

However, the 10 differentially expressed transcripts (including one pseudogene, AL731559.1) have an average abundance greater in the Mortality group patients falling above the median age of 61 compared to the ones below that age group **Figure 3** (included as **Supplementary Figure S2A**). To check for its possible modulation due to age, we also performed the simple linear regression analysis between the age and expressions of the 10 differentially expressed transcripts between Mortality and Mild patients.

Importantly, we do not see a significant association between the transcripts and the age of the patients. This suggests that despite age being an important component, it is not a major confounder alone. But in conjunction with comorbidities, treatment regimen, host immune response and the viral strain, it may affect the expression of these transcripts. We have added a paragraph of the same in the result and discussion of the revised manuscript as below:

“Within the mortality group we compared the expression of these 10 transcripts to check if there is any association between age and the outcome, while the average expression of these transcripts (excluding pseudogene AL731559.1) were more in mortality patients above the median age of 61 compared to below (Supplementary Figure S2A), there was no significant association between age and expression of these transcripts.”- Page no. 9,lines 202-207.

“As age can modulate disease severity, we compared it between the patient sub-groups. While the median age varied between mild and moderate/severe/mortality patients, there was no significant

differences between other severity groups, viz. moderate vs severe, severe vs mortality, and moderate vs mortality. Thus, we checked the effect of age on transcriptome between mild and mortality, however, found no significant association between age and the significantly expressed transcripts. This suggests that despite age being an important component, it is not a major confounder alone. But in conjunction with comorbidities, treatment regimen, host immune response and the viral strain, it may affect the expression of transcripts. Next, we compared the viral clade between different severity/outcomes and observed that despite different clinical severity, the underlying viral clade was similar (19A, 20A, 20B) (Figure 1F).” -Page no. 16, line no. 358-368.

Figure 2: Age distribution between mild, moderate, severe and mortality patients.

Figure 3: Average abundance distribution of mortality patients divided based on their median age. Patients above 61 years of age and patients below 61 years of age. It does not include the pseudogene, AL731559.1.

Reviewer #5 (Public repository details (Required)):

If the manuscript is accepted the transcriptome data from sequencing of 125 hospital admitted COVID-19 patients needs to be submitted on GEO datasets.

We thank the reviewer for this suggestion.

The raw sequencing fastq files as well as the metadata are uploaded and available at the NCBI Sequence Reads Archive (SRA; <https://www.ncbi.nlm.nih.gov/sra/>) under BioProject accession number **PRJNA678831**. The RNAseq QC files are available at (<https://doi.org/10.5281/zenodo.7471319>). The GISAID ID of the consensus fasta submitted to GISAID-EpiCoV (<https://www.gisaid.org/>) are mentioned in the Supplementary table S1.

Reviewer #5 (Comments for the Author):

The current study reveal a novel mechanism of alternative splicing in COVID-19 differential studies. The importance of transcriptomic analysis and possible mechanism of action is very well highlighted by this study. However, I have a few minor questions that you need to address.

We thank the reviewer for appreciating the findings presented within the manuscript elucidating an unexplored aspect of COVID-19 for granular understanding of the COVID-19 differential disease severity types.

1- Did you validate using another pathways tool whether you observe consistent finding?

We thank the reviewer for this suggestion.

Based on your suggestion, we have used reactome pathways and performed gene set enrichment analysis using fgsea package in R. Taking a cutoff of p value<0.1, we find similar pathways such as integrin cell surface interactions pathway, extracellular matrix organisation pathways to be negatively enriched in the mortality patients which is consistent with the pathways obtained by the KEGG database, where we find suppression of cell-cell adhesion regulation as well as integrin-mediated signalling pathways as shown in **Figure 4** below. We have included the same in the results and the figure is included as **Supplementary Figure 2B**.

“While the gene set enrichment analysis reflected similar suppression of integrin cell surface interactions and extracellular matrix organisation in mortality patients, we also observe a positive enrichment of innate immune related pathways (**Supplementary Figure 2B**).”- Page no. 10, line no. 221-224.

Figure 4: Pathway enrichment analysis using reactome for 121 differentially expressed transcripts.

2- What was the concentration of AMPure beads used to purify the library?

We thank the reviewer for this query. The library was purified using AMPure XP beads, with bead to sample ratio of 1:1. We have included the same in the manuscript at line no 544-545.

3- Figure 2 labelling and text is not matching. For eg. fig. 2 (D-H) are not correctly labelled according to the text, please correct it. Fig 3 (GH) graph boundaries thickness is inconsistent to Fig (IJ), keep it consistent.

We thank the reviewer for bringing it to our notice. During the revised submission, we have now corrected the Figure legend 2 as well as made the graph thickness uniform in Figure 3 (GH).

4- What are the limitation of your study?

We acknowledge the reviewer suggestion. In the updated submission, we have added a section discussing the study's limitations. The section is as follows:

“One of the major limitations of the study is the lack of transcript-specific pathway information for understanding transcript-specific functions. Because different transcript isoforms exhibit different expression patterns, understanding the specific function of the variably spliced transcripts is crucial. Furthermore, the study is based on samples gathered from patients on the day they were admitted to the hospital. Although the samples are optimal for examining the early host response to COVID-19, a longitudinal data can help elucidate the dynamics of alternative splicing during infection.” -Page 21 lines 485-492.

July 16, 2023

Dr. Rajesh Pandey
CSIR Institute of Genomics & Integrative Biology
Integrative Genomics of Host Pathogen Laboratory
Mall Road
New Delhi, Delhi 110007
India

Re: Spectrum01351-23R2 (SARS-CoV-2 infection Severity and Mortality is modulated by Repeat-mediated regulation of Alternative Splicing)

Dear Dr. Rajesh Pandey:

Your manuscript has been accepted, and I am forwarding it to the ASM Journals Department for publication. You will be notified when your proofs are ready to be viewed.

Sincerely,

Yun Young Go
Editor, Microbiology Spectrum
